# Peripheral Instinct: How External Devices Breach Browser Sandboxes

## Abstract

Browser APIs such as WebHID, WebUSB, Web Serial, and Web MIDI enable web applications to interact directly with external devices. The support of such APIs in Chromium-based browsers, such as Chrome and Edge, radically changes the threat model for peripherals and increases the attack surface. In the past, devices could assume a trusted host, i.e., the operating system. Now, the host is a potentially malicious website and *cannot* be trusted.

We show how this changed threat model leads to security and privacy problems, up to a complete compromise of the operating system. While the API specifications list initial security considerations, they shift the responsibility to (unprepared) device vendors. We systematically analyze the security implications of external devices exposed by such new APIs. By reverse-engineering peripheral devices of several popular widespread vendors, we show that many vendors allow controlling devices via Web APIs up to reprogramming or even fully replacing the firmware. Consequently, web attackers can reprogram devices with malicious payloads and custom firmware without requiring any physical interaction. To demonstrate the security implications, we build several full-chain exploits, leading to arbitrary code execution on the victim system, circumventing the browser sandbox. Our research shows that browser security should not rely on the secure implementation of third-party hardware.

**ACM Reference Format:**
Anonymous Author(s). 2025. **Peripheral Instinct: How External Devices Breach Browser Sandboxes. In** *Proceedings of Proceedings of the ACM Web Conference 2025 (WWW '25), (WWW '25).* ACM, New York, NY, USA, 12 pages. https://doi.org/10.1145/nnnnnnn.nnnnnnn

## 1 Introduction

Over the past years, the web has become a central platform for applications. Web browsers are among the most essential applications for end users. With a wide range of web-based applications, including webmail and entire office suites, users can accomplish many tasks purely from the browser. To facilitate these tasks, browsers introduced a range of APIs allowing websites to access various functionalities. Hand-in-hand with the APIs, browsers also rely on permissions for several of these APIs. Permissions protect users from allowing websites to perform unexpected actions. While several APIs can compromise confidentiality (e.g., webcam, or microphone) when a user grants the permission, closing the browser (or

tab) typically mitigates the security problem. For a long time, no permission has had permanent effects beyond the browser session, preventing any lasting harm to the integrity of the system. While most APIs provide functionality on a relatively high level, some new APIs have started providing direct access to devices. Such web APIs include WebUSB [1], WebHID [2], Web Serial [3], and Web MIDI [4]. Although the APIs are still experimental, they are already implemented in Google Chrome and enabled by default. Consequently, these APIs are also available in Microsoft Edge, Opera, and Electron, which rely on the Chromium engine.

In this paper, we show that adding APIs for accessing peripheral devices (e.g., WebHID, WebUSB, Web Serial, Web MIDI) has changed the threat model. Although attackers lose access to the device when the browser is closed, they can modify the device for persistence, allowing a web attacker to infect and take control of a physical device. Before these APIs, devices could assume that the host, i.e., the OS, was trusted and benign. Hence, devices assumed that all requests stemmed from legitimate users. However, since websites can interact with these devices, the host –now a website on the Internet– is not necessarily trusted. Users, browser developers, and device vendors are not sufficiently prepared for this drastic change in the threat model. While the API specifications list security considerations, they merely shift the responsibility of maintaining security to device vendors, many of which have not prepared their devices accordingly, as our analysis shows. Moreover, users are not sufficiently aware of the threats induced by providing access to peripherals. Current permission prompts, such as those indicating a website "wants to connect to a HID device" in Chrome, do not adequately convey the full security threats behind granting such permissions. This underscores the need for scrutiny before wider deployment of new specifications, as seen with the abuse potential of the new File System API [5]. Concretely, we ask the following research question in this paper:

*Which security and privacy implications does the universal deployment of peripheral access have, considering popular peripheral devices and state-of-the-art defenses?*

To answer this question, we analyze implemented Web APIs that allow low-level access to peripheral devices. Our analysis reveals two new attack vectors that circumvent the isolation guarantees of browser sandboxes, undermining the system's integrity. First, attackers can replace the entire firmware of peripheral devices from the browser, allowing them to repurpose a device. For example, a non-input device can be maliciously repurposed as a keyboard, allowing attackers to inject arbitrary keystrokes into the system. We analyze WebUSB, demonstrating that we can flash custom firmware on peripheral devices, such as the blink(1), completely overtaking and repurposing the device. Similarly, we analyze Web MIDI, showing that we can flash firmware on MIDI devices to repurpose them for malicious use cases. Second, attackers can abuse the existing functionality of devices, exploiting the shifted threat model where

the host is now malicious. For example, attackers can program custom key sequences onto a mouse button and inject them into the system whenever the victim presses the respective mouse button. The most impactful results are from WebHID, where we analyze keyboards and mice as prime examples of HIDs. We cover 22 device models from 15 vendors. All these devices allow changes to persistent settings, e.g., key mappings and macros, allowing attackers to inject arbitrary keys from outside the browser sandbox. Additionally, using Web Serial, we show that websites can send text messages via a SIM card, which can often be found in business laptops, reviving dialers.

We build multiple exploit chains to get arbitrary code execution on the victim system, entirely circumventing the browser sandbox. We show that attackers can reprogram the firmware of non-input devices and macros of input devices directly from the web. This reprogramming is mostly persistent, even surviving a re-connect of the device. We show that our attack is especially impactful on input devices, even if these devices only feature minimal macro functionality.

Our attack scenario for the Web-based attacks resembles BadUSB [6] while reducing requirements and limitations. We do not rely on known bugs or vulnerabilities to exploit USB devices [7, 8] and do not need to attach custom USB devices to the victim machine [9]. We solely rely on the well-defined and voluntarily-provided interface of the already-attached device. When reprogramming macros, our attack does not change any device or device descriptor from the operating system's perspective, circumventing software [10–13] and hardware [14, 15] solutions to prevent BadUSB attacks. Even worse, as soon as a user allows a website to access peripherals, our attack can be mounted by a malicious website directly from the browser without requiring any download or browser vulnerability. Consequently, countermeasures against such an attack are challenging without sacrificing functionality.

To mitigate peripheral-based attacks, we advocate for an additional opt-in mechanism on the device that is honored by the browser. Vendors could use it to indicate that they are aware of the changed threat model of the APIs allowing peripheral access. As a result, browsers would know it is safe to expose this device to the web. We implemented our proposal as a proof of concept in Chromium and a programmable open-source HID. We show that only minimal changes are required to Chrome, with a total patch of a single line of code. Most importantly, our proposal is fully compatible with the HID specification and does not break existing functionality.

While minor changes to devices and browsers would make peripheral-based attacks more complex, we show that exposing devices to the web poses a significant threat. Our research demonstrates that the new threat model with a malicious host that these Web APIs introduce is often not considered during the design of peripherals. Given the wide distribution of Chromium-based browsers and affected peripheral devices, we consider this problematic as it affects a large user base. We hope our insights raise awareness among device vendors, leading to more secure devices. Moreover, our research shows that browser security should not rely on third-party hardware vendors alone, and novel web APIs cannot simply shift the responsibility to third parties without compromising security.

**Contributions.** We summarize our contributions as follows.

- We analyze the attack surface of Web APIs that provide low-level access to peripheral devices.
- We present novel attacks on the host via browser-based APIs, entirely circumventing the browser sandbox and leading to arbitrary code execution.
- We demonstrate systemic improvements to the security of the vulnerable API specifications.

**Responsible Disclosure.** We have received positive feedback from Chromium and Logitech. The Chromium team collaborates with device vendors to implement some of our recommendations, such as the requirement for physical user interaction during reprogramming. They also aim to improve the clarity of permission prompts. Logitech acknowledged our findings and plans to implement our proposal of configuration functionality on a separate usage.

**Availability.** We open source all our experiments and proofs-of-concept on acceptance of the paper.

## 2 Background

In this section, we cover relevant device protocols, including current implementations as Web APIs and known security considerations.

### 2.1 Device Protocols

Devices connect to a host system via various protocols. Some protocols are general-purpose carrier protocols, such as USB or Serial, while others are specialized protocols, such as HID or MIDI.

**USB & Serial.** *Universal Serial Bus* (USB) is a wired communication protocol for high-speed data transfer and power delivery. It is the de facto standard for connecting peripherals to computers. USB is a general-purpose protocol that supports various device classes, such as Human Interface (HID), Mass Storage, and Audio Devices.

A *serial port* is a communication interface that transfers information sequentially, one bit at a time. Serial ports have been commonly used in personal computers to transfer data to devices such as modems, terminals, peripherals, and between computers. While USB has largely replaced serial ports, they are still used to control embedded systems or interact with legacy devices (e.g., modems).

**HID & MIDI.** The *Human Interface Device (HID)* protocol is a standardized protocol for communication between devices and hosts. It is transmitted via USB or Bluetooth and used for devices to interact with a human operator [16]. Data packets exchanged between the host and device are called *HID Reports* [17]. Devices send *input reports* to the host, while hosts send *output reports* to the devices. Additionally, there are bi-directional *feature reports* intended to configure a device. *HID report descriptors* describe the binary format of reports supported by a device and can be enumerated by the host [17]. The *Usage* describes the intended use of a device and the purpose of reports. OSs ship with default drivers for many standardized classes of HIDs, such as keyboards and mice [18]. A *Musical Instrument Digital Interface (MIDI)* allows electronic musical instruments and computers to communicate and synchronize [19]. It can operate over various transport protocols, including USB and Bluetooth. Data packets, known as *MIDI messages*, carry musical parameters (*channel messages*) or system settings (*system messages*).

## 2.2 Device Browser APIs

The web platform already supports input from some device types (e.g., HIDs) via OS drivers that provide an abstraction of a device [2]. However, some devices lack OS support (e.g., gamepads). Furthermore, device-specific communication or configuration is not exposed to web pages. Device Browser APIs bridge this gap [2] and allow web pages to communicate with devices directly.

### 2.2.1 Browser Support.
Currently, five device APIs are available in Chromium-based browsers: WebHID, Web MIDI, Web Serial, Web Bluetooth, and WebUSB [20]. The respective API standards are currently drafts at the World Wide Web Consortium (W3C). They have not received formal reviews and are not endorsed by the W3C, and thus no official standards [21]. Still, the APIs are implemented and enabled by default in Google's Chrome for desktop and Chromium-based desktop browsers [20], including Microsoft Edge and Opera since March 2021. Together, they account for over 80 % of the desktop browser market share [22]. ChromeOS, which leverages Chrome as its application platform, has deprecated its previous APIs (e.g., `chrome.hid`) and now requires user-space apps to use browser APIs for device access [23]. Both Mozilla and Apple do not implement these APIs at the moment [24, 25], with the exception of Web MIDI in Firefox which is implemented behind a *site permission add-on* [26].

### 2.2.2 Security Considerations.
Chrome uses various security measures to protect the host, device, and user. As the API standards are not yet finalized, the following measures apply to the most recent version of Google Chrome (121) at the time of writing.
**TLS only.** Device Browser APIs are only available in a secure TLS context to prevent MITM attackers from accessing the API [27].
**Blocklist.** According to the proposals, each API, except for the Web MIDI API, includes a blocklist [28–31]. Depending on various properties and identifiers, devices may be blocked and hidden. The specifics of the respective blocklists are discussed in Appendix A.
**Permissions & User Activation.** The standard proposals recommend implementing a chooser-based dialog with at least two clicks (to reduce the possibility of accidental clicks) for requesting device access [1, 2]. The only exception here is the Web MIDI API, which only has a single click prompt [4]. The initial permission dialog is guarded by transient user activation [32, 33]. Permissions persist until the user or site explicitly revokes them.
**Recommendations in API Proposals.** Although the API proposals have mentioned potential security risks since at least 2019 [2], our investigation shows these considerations are largely unaddressed. This indicates the need for research to raise awareness of the API and its capabilities. Our work underlines that the W3C underestimates the prevalence and impact of device-associated threats. For example, while the W3C acknowledges that an HID "may contain [...] programmable macros" and suggests that "device manufacturers must [...] prevent a malicious app from reprogramming the device", we show that the issues remain unaddressed.

## 3 API Security Analysis

In this section, we provide an overview of the security implications of device APIs due to the drastic change in the threat model. We bootstrap our evaluation on APIs included in the Permissions Policy [34, 35], following the assumption that all critical APIs must be gated by a prompt. Out of those, we focus on APIs that interact with peripheral devices. For all remaining APIs, we manually evaluate if they allow attackers tampering with the devices. Following these steps reveals five browser APIs: Bluetooth [36], WebHID [2], Web MIDI [37], Web Serial [38] and WebUSB [1]. In the remainder of this work, we do not expand further on the Bluetooth API, as the implications of the API are analogous to those of other transport protocols such as USB.

## 3.1 Threat Models

The threat model of devices traditionally only spans the host system and the device itself [39]. This model only allows for two directions of *local* exploitation. First, a device can exploit vulnerabilities on the host, e.g., by injecting keystrokes to execute commands or extract sensitive information. Second, a host can exploit vulnerabilities on shared devices (e.g., printers) or security hardware (e.g., FIDO security keys) to extract confidential data. With device APIs in browsers, a new threat model emerges in which the devices are *exposed to third parties* via browsers. Here, the device may process malicious requests from an—in the classic threat model—trusted host system, which forwards the API-initiated requests of an untrusted site. This confused deputy attack enables several devastating security-critical attacks, as shown in Section 3.2. The only assumption is that an attacking website gains user permission for the respective Device API, for which there are several ways (cf. Section 3.3).

## 3.2 Attacks Enabled by Device APIs

Using a device API, a malicious website can send data to a device. Below, we discuss the security implications of this new threat model.
**Integrity.** The focus of our investigations are threats to device and system integrity. First, many devices allow modifying or replacing the firmware via an exposed bootloader (Section 4) enabling BadUSB-like attacks [6]. The capabilities of an attacker depend on the capabilities of the device (e.g., Bluetooth) but generally allow emitting trusted input events via HID. Second, as shown in Section 6, devices can be "reconfigured" or controlled *without* replacing the firmware. For example, several mice or keyboards allow users to reprogram buttons with macro functionalities, all of which are exposed via WebHID. A malicious actor can abuse this to escape the browser sandbox. Similarly, modems are accessible via Web Serial, allowing attackers to control the modem (see Section 6).
**Availability & Confidentiality.** The device API proposals also state some other concerns only discussed briefly in this work. A malicious actor could perform Denial-of-Service (DoS) attacks on the device to temporarily or permanently disrupt the functionality of the device. Similarly, device APIs may violate confidentiality. For example, macro features in keyboards may be used to quickly enter sensitive information such as credentials or passwords[1]. Device APIs potentially enable web attackers to extract such sensitive data by reading the on-board device storage.

---

[1]https://security.stackexchange.com/q/222210

### 3.3 Gaining Device API Permission

To launch any attack leveraging device APIs, malicious websites must gain access to the respective API. Attackers have several ways of obtaining such privileges. In all cases, the user must grant permission to use a device on some site. Once permitted, the site can interact with the device without further consent on future visits. In the simplest scenario, the user grants permission to a malicious website directly. Attackers may leverage social engineering techniques (i.e., phishing) to trick users into granting permissions, e.g., by impersonating legitimate vendor sites. For example, Hazhirpas et al. [40] convinced up to 95 % of users into granting permissions leveraging a browser game. Furthermore, an attacker can leverage permissions granted to another site via a Cross-Site Scripting (XSS), website compromise, or domain re-registration [41]. XSS in particular is one of the most prominent security issues for websites and frequent in the wild [42]. In addition, browser extensions can manipulate requests and execute JavaScript in the context of arbitrary sites [43]. As such, malicious or vulnerable browser extensions can also be used to gain access to devices. Lastly, users of rehosting services may inadvertently grant permissions to a malicious site since such services frequently merge the origin of all proxied sites [44].

## 4 Firmware Attacks

Devices commonly implement firmware update mechanisms. This is usually implemented as a bootloader that allows flashing new firmware. Communication with the bootloader is often done via the same interface as the device itself, e.g., HID or MIDI. Such mechanisms are implemented on most devices regardless of their transport protocol. In this section, we analyze the risks of exposing firmware update mechanisms to web-based APIs. We identify two primary attack vectors, namely, allowing the flashing of custom firmware (Section 4.1) and firmware rollbacks (Section 4.2). We illustrate attacks on two different web APIs via case studies on the Logitech Unifying Receiver (WebHID/WebUSB) and the Launchpad MK2 (Web MIDI). However, this class of attacks is not limited to the two presented APIs but rather an overarching issue with exposing firmware update mechanisms.

### 4.1 Custom Firmware

Allowing the host to flash custom firmware onto a device via a browser-based API is a severe security risk. Custom firmware can re-program the device to perform almost arbitrary attacker-controlled functionality. In the following, we present two case studies that show the practicality of this attack vector.

**Logitech Unifying Receiver.** The Logitech Unifying Receiver is a proprietary USB wireless receiver based on transceivers of the nRF24L-family used for a wide range of wireless keyboards and mice from Logitech. The wireless receiver communicates using the custom HID++ protocol (see Section 5.1.3) and features an HID-based bootloader that allows replacing the firmware. As a response to a variety of vulnerabilities reported by Bastille Research [45, 46] (e.g., CVE-2016-10761) and Markus Mengs [47] (e.g., CVE-2019-13053), Logitech introduced signed firmware updates in 2019 (i.e., RQR12.09 and RQR24.07). However, wireless receivers sold with older firmware, i.e., firmware before 2019, allow flashing arbitrary firmware unless a user manually updates the device. A custom

firmware, for example, allows emitting trusted input events such as keystrokes. Such a compromised device fully covers the functionality of USB Rubber Ducky devices [9] (see Section 6.1). This effectively allows an attacker to execute arbitrary commands on the host system. Additionally, the device provides a malicious actor access to the 2.4 GHz transceiver module, which can interact with other devices using the Enhanced ShockBurst protocol. Access to the transceiver module allows, for example, abusing the MouseJack vulnerability [46]. We verify that we can flash the firmware used to exploit the MouseJack vulnerability using WebHID onto a Logitech Unifying Receiver. With the custom firmware, we successfully inject arbitrary keys into a different laptop in the same room that uses a Logitech MX Anywhere 2S wireless mouse. This attack only requires granting the WebHID permission to exploit the wireless receiver and no user interaction by the ultimate target user. Thus, control over a wireless receiver running outdated firmware can be used to compromise systems in the proximity of the victim device.

**Launchpad MK2.** The Launchpad MK2 is a MIDI controller widely used in music production. Below, we analyze its firmware update mechanism. The Launchpad can be forced into bootloader mode using a specific SysEx message [48]. Firmware updates are provided as *syx* files (files containing SysEx messages). They can be applied by "playing" the file (i.e., sending the SysEx messages to the device) when the Launchpad is in bootloader mode. Using the open-source toolkit to build custom firmware for a similar Launchpad [49], we reverse-engineer the SysEx message format and extract the raw firmware binary. The firmware comprises a full ARM image for an Arm Cortex M4, enabling unconstrained privileged code execution on the device. While the Launchpad MK2 appears as a MIDI device, it is connected to the PC via USB and uses USB as a carrier protocol. Thus, with complete control over the executed code, it is possible to modify the low-level USB implementation to implement an HID device instead. We successfully patch the firmware achieving arbitrary code execution on the device.

### 4.2 Firmware Rollbacks

While devices may not allow flashing arbitrary firmware by enforcing vendor signatures, they may still be vulnerable to firmware rollbacks. Such a rollback can bring the device into a state where old, vulnerable firmware is installed, which can then be exploited. Such an attack is possible with up-to-date Logitech Unifying Receivers. Our investigation shows that they do not implement *rollback protection*. This protective mechanism prevents downgrading the firmware to older versions [50]. For example, we can construct a full-chain exploit for a known and fixed vulnerability (CVE-2019-13055). Our target device was the Logitech Unifying Receiver C-U0008, running the latest firmware (RQR24.11). The vulnerability allows extracting AES keys to encrypt the receiver's wireless traffic and a wireless device (e.g., keyboard) via USB/HID. Our full-chain exploit first restarts the wireless receiver into DFU mode using a specific HID output report and then flashes the older firmware version using HID output reports. Afterward, we extract the AES keys as described by the original vulnerability. Note that the DFU mode of the wireless receiver reports as a different device, such that a malicious actor must obtain two separate permissions. This process takes approximately 38 s.

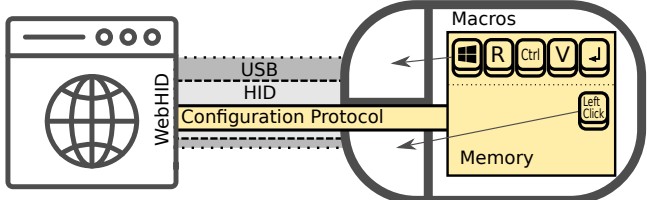

Figure 1: **HIDs (right), such as mice, communicate with the browser (left) using WebHID. Custom configuration protocols are built on top of the HID protocol on top of USB. Programmable devices feature on-board memory to store macros for keys/buttons in a custom macro language.**

## 5 Analysis of Device-Specific Protocols

While firmware attacks are a severe threat and grant a malicious actor full control over a device, they have several prerequisites. Alternatively, the attacker can abuse the device's existing functionality to perform attacks. In contrast to firmware attacks, such attacks differ from device to device and require a detailed understanding of the device. In this section, we first analyze the macro functionality often found in HID devices. This functionality allows users to reprogram keys or buttons to perform custom actions and is implemented differently across device vendors and devices. Second, we discuss how modems can be controlled via a serial interface using the AT command set.

### 5.1 HIDs with Onboard Macros

Figure 1 provides a high-level overview in which an HID is exposed to WebHID and allows attackers to use the configuration protocol to change macros stored in memory. In the following, we study each of these aspects in more detail on a subset of devices representative of the implementation choices of device vendors. We rely on these results to instantiate macro-based attacks in Section 6.

*5.1.1 Overview of HIDs and Methodology.* Table 1 in Appendix B provides an overview of the devices discussed in the following. We investigate the features of 22 devices from 15 vendors, covering many widespread devices from large vendors, such as Logitech and Microsoft, as well as from smaller vendors. We are mainly interested in reprogrammable on-board macro functionality supported by 14 devices. Further, we investigate to which extent this functionality can be accessed from WebHID. Since our Razer devices only expose this functionality via USB and not via the HID protocol, they are not discussed in detail. For our analysis, we rely on protocol reverse engineering using USB dumps, prior work on protocol reverse engineering, and documentation. We use Wireshark's capability of dumping USB traffic [51] on a laptop that does not connect any internal devices using USB to filter relevant packets. Using the official closed-source tools, we analyze the effects of different settings on the raw bytes sent via USB to understand the protocol.

*5.1.2 Macro Capabilities.* Generally, a macro consists of a sequence of actions, such as key/button press/release or actions to control a mouse pointer. Moreover, many devices allow custom delays between actions. The analyzed devices show vast differences in macro capabilities, ranging from a single keycode to complex macro languages. We group the macro capabilities into 3 categories.

**Single Key.** The simplest macro implementations only allow defining a single keycode that the device sends when the corresponding button is pressed. Such macro functionality is the least critical from a security standpoint, as the exploitation is severely limited. However, it still is, e.g., possible to build a wiretap on an attacker-controlled site using an auto-focussed hidden input field and the key combination that activates voice typing (cf. Section 6.2). We find such an implementation on the Microsoft Pro IntelliMouse.

**Key Sequence.** The majority of analyzed devices support keystroke sequences of varying lengths. In the simplest case, these sequences have a fixed upper bound for the length (e.g., 5 for the CH57x). For other devices, e.g., from VIA or Logitech, the limiting factor is the available on-board memory. In addition, the devices differ in how fast they replay macro actions. We provide the maximum length and the minimum time between keystrokes in Table 1 in Appendix B. Key sequences also come in different encodings. The most simplest encoding is a sequence of HID keycodes. This encoding does not require any complex logic on the device (e.g., CH57x). If other functionalities such as mouse movement (e.g., Zelotes) or custom delays (e.g., VIA, Zelotes) are supported, lists of custom structures are used for storing the macros.

**Instruction-set Emulators.** Logitech uses the most complex and flexible encoding of all analyzed devices. They store macros as custom variable-length instructions that are interpreted by an instruction-set emulator. The devices support a wide variety of instructions that range from delay, key press, and mouse functionality instructions to sophisticated control flow instructions (e.g., 'unconditional jump' or 'jump if the macro button is released').

*5.1.3 Proprietary Configuration Protocols.* 14 of our tested devices use custom protocols built on the HID protocol. These protocols range from straightforward protocols using 4 HID reports to program a key to complex protocols requiring 800 HID reports to achieve the same. Most devices feature error-tolerant protocols and firmware. Logitech, e.g., uses a default profile if the profile information in memory cannot be parsed. The Logitech and the Zelotes software reset the device to factory defaults if the information cannot be parsed. While several protocols are documented or have been (partly) reverse-engineered [52–54], we provide details of our findings for the Zelotes and CH57x protocol in Appendix F.

### 5.2 Hayes and Hayes-compatible Modems

In this section, we discuss how modems can be controlled via the Web Serial API. We focus on the AT command set, also known as the Hayes AT command set, which is a standard for controlling modems that was introduced in 1981. Modems that support the AT command set are called Hayes-compatible modems. An example of a Hayes-compatible modem is the *Fibocom L850-GL* LTE modem. It is a PCIe modem used in various laptops, such as the Lenovo ThinkPad X1 Carbon or the Lenovo ThinkPad T490s.

The AT command set is text-based and uses ASCII characters [55]. The commands are sent to the modem via a serial interface, such as UART or USB. Similarly, the modem also responds using plain text. AT commands are used to communicate with different services

provided by the modem. Commonly, AT commands are used to control call services, cellular network services, and SMS services [55]. Note that some vendors may extend the AT command set with proprietary commands. For example, `AT+CMGF=1\r\n` instructs the modem to switch to text mode for SMS messages. In the following, `AT+CMGS="+00123456789"\r` defines the recipient of an SMS message. Lastly, the message content is defined and sent using `Hello World\x1A`. Section 7 demonstrates how this API can be used for malicious activities.

## 6 Device-specific Attack: Exploiting Onboard Macros

The first device-specific attack vector targets macro-capable devices to inject keystrokes into the host system. Our investigation in Section 5.1 revealed that 14 of the 22 tested HIDs allow reprogramming over WebHID, including vendors such as Logitech and Microsoft. In this section, we demonstrate resulting threats, leading to browser-sandbox escapes via key injections and exposure of potentially confidential information via key injections.

### 6.1 Command Injection

Our first attack method aims to compromise a target system by circumventing the browser sandbox. To this end, the web attacker reprograms a HID with a critical chain of commands that spawn an attacker-controlled program. For brevity, we focus on Windows systems, as they are the most prominent target system [56]. Other systems, such as macOS and Linux, are discussed in Appendix D. While the attack is similar to the well-known Rubber Ducky attack, the device is merely reprogrammed and not fully under the attacker's control. This imposes several limitations on the attack, such as the inability to query host system information or even a strict limit on the number of keystrokes that can be injected.

Keystroke injections all follow the same basic principles. At first, a sequence of keystrokes provides the attacker access to a *run dialog*, a part of a system that facilitates dynamic code execution. The remaining part of the injection provides inputs to the run dialog and contains the actual payload executed on the system.

**Run Dialogs.**  There are multiple run dialogs on Windows. The Start menu can be accessed using a single keystroke, `⊞`. Alternatively, the Run command window can be accessed using `⊞`+`R`. Both run dialogs provide similar functionality. They may be used to navigate to web resources using the system's default browser, to start applications, or to run arbitrary commands. Another run dialog is the Quick Link menu (`⊞`+`X`), which can open the Windows Powershell as Administrator using only five key presses. If the user is logged in as an Administrator, this only requires confirming a dialog which is possible using `◀`, `↵`. Using keystroke combinations and commands that do not affect one OS but perform actions on another OS, we may build a polyglot injection that works regardless of the victim system. As an example, a polyglot run dialog opener for Windows and Linux can be achieved using the following keystrokes: `⊞`+`X`, `I`, `Ctrl`+`Alt`+`T`.

**Injection Payloads.**  Keystroke-based payloads are well explored, mainly due to the *USB Rubber Ducky* [9], a programmable USB device designed for keystroke injection attacks. An extensive library of payloads for the device [57], covers most popular attack targets. Our injection payloads, however, have more limitations, as discussed previously. For example, less than 15 % of payloads from the official repository would work on the Logitech G203 due to the limitation of 80 keystrokes. A simple downloader payload that downloads and executes a script from an attacker-controlled website amounts to approximately 50-60 keystrokes. To significantly reduce the number of keystrokes, attackers can store an ephemeral payload in the system-wide clipboard.

Write-access to the clipboard from the browser is only guarded by transient user activation [58] and thus implicitly granted with the WebHID permission. Leveraging the clipboard, the shortest payload is `Ctrl`+`V`, `↵`. Thus, the shortest total injection length amounts to 3 keystrokes on Windows. Such an injection can be performed using all programmable devices discussed in Section 5, except for the Microsoft Pro IntelliMouse. Using the Keychron V1Z2, the injection only takes about 35 ms. Similarly, the shortest injection with elevated privileges uses 7 keystrokes on Windows and takes about one second using the Keychron V1Z2.

**Time to Configure.**  Our investigation shows that the time to configure is well below 1 s for all devices in our set. Appendix B includes a table of the times for a subset of devices.

### 6.2 Spyware

Besides the command injection, which aims to gain remote access to a system or exfiltrate sensitive data, keystroke injections via macros can also be leveraged to spy on a user's behavior. Most intuitively, an attacker can log user activity using screenshots. Here, they assign a frequently used button to a macro that first issues a key combination that triggers a screenshot and then performs the expected behavior to remain stealthy. On Windows, e.g., the `PrtScn` (i.e., *PrintScreen*) key captures a screenshot of the entire screen and stores it in the clipboard [59]. It can be exfiltrated (using `Ctrl`+`V`) or via the clipboard history. Further, it is, e.g., possible to construct an audio wiretap. On Windows, a payload may use the keyboard shortcut to activate the in-built dictation tool, `⊞`+`H`, to build a wiretap [60] that writes all spoken words to the current cursor location. Exfiltration can, e.g., happen via an attacker-controlled website with hidden input elements. Note that this does, however, trigger a message display by the OS and a sound.

## 7 Device-specific Attack: Exploiting Hayes-compatible Modems

As introduced in Section 5.2, Hayes-compatible modems can be controlled via the AT command set. In the following, we discuss several potential threats that can be exploited by a malicious actor that can control a modem via the Web Serial API. The threats range from dialing or sending SMS to premium-rate numbers, over GPS tracking to permanent denial of service attacks. These attacks assume that a SIM card is inserted into the modem, which commonly occurs in a business setting. To prevent unauthorized usage of a SIM card, usage is commonly gated by a PIN code. Upon booting, the system prompts the user to enter the PIN code.

**Reviving Dialer Campaigns.**  In the past, a dialer was a common type of malware that would dial premium-rate numbers, resulting in a charge to the victim [61]. The attacker would set up a premium-rate number, which features high charges for the caller. The dialer

malware would then dial the number, resulting in a charge to the victim. In recent years, dialer campaigns have become less common due to the widespread use of broadband internet connections. However, the Web Serial API allows reviving dialer campaigns. Given serial access to a modem, an attacker can issue AT commands that dial or send SMS to premium-rate numbers (see Section 5.2).

**Spyware.** The modem contains various privacy-relevant information that can be accessed via AT commands. For example, many modems contain a GPS module that allows the modem to determine its location, which allows tracking a user's location. In addition, the modem also contains information about recent activity, such as dialed numbers or SMS messages. SMS messages, in particular, can contain sensitive information such as two-factor authentication codes or passwords. Such information can even be intercepted by forwarding SMS messages and calls to the attacker's number.

**Permanent Denial of Service.** If the SIM card is not unlocked, a malicious actor cannot use the modem for the advanced attacks discussed in the previous sections. To hinder brute-force attacks, SIM cards are locked after a certain number of incorrect attempts [62]. Commonly, after three incorrect attempts, the SIM card is locked, and the user has to enter a PUK code to unlock it. Then, after some incorrect attempts with the PUK code, the SIM card is permanently locked, and the user has to be issued a new SIM card by the carrier. PIN and PUK are entered using AT commands, which allows an attacker to perform a permanent DoS attack locking the SIM card.

## 8 Mitigating Device API Attacks

In this section, we propose and discuss mitigations for Device API attacks. While disabling the APIs mitigates the underlying issue, this is a drastic measure. We discuss the advantages and disadvantages of other mitigation approaches that are fully backward-compatible with the respective standards and ideally also compatible with the current implementation of the APIs.

### 8.1 Extension-based Control

The extension-based approach introduces an abstraction layer between the low-level access to the device and the interface exposed to any (malicious) site. This layer can inspect the data sent to a device, which allows the extension to implement a firewall-like mechanism to prevent arbitrary sites from performing security-relevant actions, such as macro programming or firmware updates. Alternatively, the extension can also completely prevent access to low-level functionality and instead expose abstract functions that encapsulate the device's behavior. Such an abstract function could, e.g., allow any site with access to change the LED color of the connected device (e.g., `device.setRGB(0,255,255)`) without exposing any other functionality. Such an extension could be provided by the device vendor that protects access to the device against malicious use by restricting access to sensitive functionality. It would even require fewer privileges than regular native configuration software, which may achieve full control over the OS, as it can, for example, emulate keystrokes directly. Similarly, it is possible to implement an extension that blocks sending data or even removes an API completely.

**Implementation.** The mechanism is implemented by leveraging a technique called *Virtual Machine Layering* [63, 64], which allows intercepting JavaScript functions. Here, the API functions that can be used to send data to a device are encapsulated by functions that first filter their arguments and then pass those arguments to the device via a handle to the original function. The original function can only be called using the handle in the encapsulation. A Chrome Extension with Manifest Version 2 can use the manifest in Listing 1 in the appendix to run a content script before the rendering of any arbitrary site (and subdocuments) injects the encapsulation before the head of the document. After the initial encapsulation, any script that tries to access the WebHID API can no longer access the original WebHID API. As a proof of concept, we implement a Chrome Extension that prevents a site from accessing the onboard memory feature on Logitech devices relying on the HID++ 2.0+ protocol via the WebHID API (cf. Section 5.1.3). This is easily possible since the third byte of a report is the feature identifier.

### 8.2 API-Device Contracts

The WebUSB proposal discusses the possibility of instantiating API-device contracts that are honored by the browser [1]. While such contracts are an impractical solution for the broader set of USB devices due to the lack of standardization, they may be feasible for a subset of devices. In the following, we propose two approaches to implement such contracts for WebHID with minimal impact on devices that aim to support the API. HID Usages allow grouping related controls by specifying a Usage Page and a Usage ID associated with a report, forming the so-called 32-bit extended Usage.

**Reserved Usages.** We propose an approach where devices implement security-relevant functionality, such as persistent memory manipulation, either behind a reserved Usage Page or a reserved Usage ID. For most firmware, this only requires minor firmware modifications. To evaluate the feasibility of this approach, we modify the firmware of the blink(1) [65], an open-source device that supports WebHID and has a driver integrated into the upstream Linux kernel, to expose less functionality via WebHID. It uses two feature reports operating under the same Extended Usage. The feature report with ID '2' exposes the "reboot to bootloader" functionality. To separate this functionality, we introduce a feature report with ID '3' that operates under Usage Page '0xdead'. The code of the official tool for the device has to be modified by a single digit. We extend the Chromium blocklist locally to prevent any reports associated with Usage Page '0xdead'. Only reports under the blocked Usage Page can no longer be sent or received.

**WebHID Capability Indicator.** The WebUSB specification proposes a WebUSB Platform Capability Descriptor implemented by storing a specific Platform Descriptor in the Binary Object Store of the USB device [66]. This descriptor provides basic information about the device to the browser and could also be used to indicate that a device should be exposed via WebUSB. Similarly, a report under Usage Page '0xdead' could indicate that the device acknowledges the WebHID threat model. Since this Usage Page is not standardized, the device is free not to implement any functionality on this report and only use it as a flag.

**Browser-Device Communication.** The WebUSB specification discusses two other approaches to mitigate attacks on USB devices [1]. The first mechanism is similar to the *Referer* header [67] and would be used by the browser to communicate the origin from

which a request originated to the device. This mitigation, however, shifts the responsibility of access control to the device. Meanwhile, the second mechanism is similar to *Cross-Origin Resource Sharing* [68]. Here, a device can specify a set of origins allowed to communicate with the device. The browser then enforces this allowlist. The proposal states that such an allowlist need not be implemented on the device but could also be specified in a separate public registry. Such an approach may be challenging to maintain as it requires maintenance of the registry or modifications to firmware.

### 8.3 Pure Browser-based Approaches

**Directional Permissions.** Bi-directional communication may not be necessary in many scenarios. Hence, splitting permissions into input and output would be possible. The default permission would only allow receiving input from the device. Permission to send data to the device could be granted separately, with an additional warning. This way, non-standard controllers could, e.g., be used in browser games without the risk of reprogramming.

**Permission Dialogue.** As of writing, the permission dialogue follows a chooser-based approach, which requires at least two clicks to grant permission for a device [2]. However, there is little to no information about what the site may achieve, given access to a device. Further, the Device API permission prompts are visually and operationally similar to the prompts by non-security-relevant browser APIs such as the notification API. Given that prior work on browser extensions [69] and Android permissions [70] established that users rarely understand the risks associated with permissions, this likely also applies to browser APIs. Moreover, inconsistent browser implementations [71] obfuscate implications for the end user. Further, Progressive Web Apps (PWAs) are web applications that can be installed on a device [72]. Since installing PWAs requires further consent and user interaction, they may be granted permissions that go beyond the capabilities of a regular website.

**Allowlist.** The several specifications and Chromium implement blocklists restricting access to devices. Vendors have to submit rules proactively. While blocklist-based approaches are often easy to manage and set up, they often grow exponentially and may still miss entries to block access to malicious devices [73]. In contrast, allowlist-based approaches usually take more time to manage but provide greater security due to the restrictive default behavior [73]. As such, an allowlist-based approach may be a better fit due to the prevalence of insecurely configurable devices.

### 8.4 Device-based Approaches

Update mechanisms are crucial to ensure that vulnerabilities are fixable by the end user. However, devices must ensure attackers cannot exploit the update mechanism. Device vendors should employ authenticated firmware update mechanisms [50]. Here, the origin and integrity of firmware updates are verified using cryptographic signatures to prevent malicious ones. In addition, to prevent firmware downgrades to vulnerable versions, devices should also employ rollback protection, which checks the firmware version before performing an update [50]. The update mechanism may also require physical interaction with the device (e.g., pressing a button) to ensure a user-initiated update. This requirement can additionally be imposed during runtime reprogramming of the device.

### 8.5 Host-based Approaches

Since USB is a prominent attack vector, a wide variety of allow- or blocklisting approaches [10–13] has emerged to deal with USB threats such as BadUSB [6]. Such an approach would, e.g., defend against BadUSB-like attacks performed via HID-based bootloaders since the firewall can prevent devices from sending input by default. However, this is ineffective against macro-based attacks since the intended functionality of a device is abused. Since the behavior and, in particular, the time between keyboard events differ from normal user behavior during keystroke injection attacks, various injection-detection approaches have emerged [74–77]. These approaches leverage timing differences between keystrokes, letter frequencies, keypress times, and latency. To our knowledge, these mechanisms are not widely deployed due to their performance costs and relatively high false-positive rates. However, our minimal injection sequences of 3 keystrokes only feature 2 data points for such mechanisms, so any detection mechanism is heavily limited.

**OS Device Access Control.** On Linux, *udev* is the generic device manager that provides an abstract interface of the hardware to the rest of the software [78]. By default, HIDs are not accessible to unprivileged users. The device manager has an extensive set of rules that allows customizing this behavior. This default behavior also prevents opening devices from WebHID on Linux systems unless Chromium runs as a privileged process or the necessary *udev* rules are present. Similarly, Apple's macOS also blocks applications from accessing devices that implement a keyboard, mouse, or trackpad to prevent input monitoring [79]. As such, a user must grant the browser permission to access such devices.

### 8.6 Recognizing Macro-exploitable Devices

In order to estimate whether a device is exploitable, we propose a simple tool that can be used to rule out a device is exploitable with WebHID. The tool is based on observations from our analysis of 22 devices Section 5. All exploitable devices implement the **Keyboard or Keypad Usage** and support **output or feature reports** under a Usage that is accessible using the WebHID API. As such, devices that lack one of these two features are definitely not exploitable. For the remaining devices, we observed that output or feature reports that carry more than 64 bytes were only used for programming macros or exchanging the memory or firmware of the device. Thus, the presence of high-capacity output or feature reports increases the likelihood of a device being exploitable.

## 9 Conclusion

We showed how Device APIs change the threat model for peripherals, leading to severe security and privacy problems. Based on our reverse-engineering and analysis of devices from several vendors, we found that many allow device control from within the browser, up to reprogramming or even fully replacing the firmware. Consequently, malicious websites can control devices without requiring any physical interaction. To demonstrate the security implications, we built full-chain exploits, leading to arbitrary code execution on the victim system, circumventing the browser sandbox. Our research highlights the need to raise awareness among device vendors, indicating that the web might not be ready yet for a global deployment of Device APIs, given their security implications.

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

## A  Accessible Device Classes

In this section, we provide a generic overview of the device classes that are accessible via the critical device APIs.

**WebHID.**  WebHID allows access to almost all HID devices, including keyboards, mice, and gamepads [2]. However, for security reasons, the user agent blocks access to FIDO U2F collections and input-related HID collections for keyboards, mice, and keypads. This prevents spoofing of communication with FIDO functionality, the creation of input loggers and circumvention of the operating system's focus model. Most notably, we show that devices with macro functionality are generally reprogrammable via additional collections in Section 5.

**Web MIDI.**  Web MIDI allows access to MIDI devices, such as electronic musical instruments, MIDI controllers, and MIDI interfaces [37]. While the OS may provide drivers for MIDI devices, the protocol itself is not used by the OS. Instead, MIDI devices are typically accessed by (proprietary) music production software. Most MIDI messages relate to musical performance and thus do not pose a direct threat, even if an attacker can send arbitrary MIDI messages. However, MIDI supports *System Exclusive* (SysEx) messages which add device-specific (non-standardized) functionality. In Chromium-based browsers, the prompt is the same, regardless of whether SysEx messages are exchanged.

**WebUSB.**  While USB is a generic protocol, WebUSB only allows access to a small subset of devices [1]. The user agent blocks access to a set of USB interface classes for which most OSs have built-in drivers, such as mass storage, HID, and audio/video devices. As a rule of thumb, every type of interface available via some other high-level API (e.g., WebHID) is blocked by WebUSB. Thus, WebUSB can primarily only access generic interfaces or devices that are not covered by other APIs.

**Web Serial.**  The Web Serial API allows access to serial devices, such as microcontrollers, GPS modules, 3D printers, and other devices that communicate via a serial interface [38]. Such devices can be connected to the host system via USB or other interfaces. Further, the API allows access to the serial ports of Bluetooth Classic devices.

## B  Macro-based Exploits

Table 2 shows the time it takes to reprogram a device. This time is measured from the user interaction that triggers reprogramming until the macro is entirely written to the device and activated by, e.g., also modifying the current active profile. Here, we measure the time it takes to program the shortest malicious macro (i.e., ⊞ + R , Ctrl + V , ↵ ), as discussed in Section 6.1. This time, however, only represents an upper bound since our web applications are not optimized for performance. Further reverse engineering efforts may drastically reduce the time to configure.

## C  Run Dialogs

Table 3 gives an overview of the various run dialogs that may be used to execute payloads, as discussed in Section 6.1. The table also shows the time to payload, which measures the time it takes until the dialog appears after the start of a macro such that a malicious actor may enter the actual payload. This time was measured using the Keychron V1Z2, which featured the smallest time interval between macro keystrokes (see Table 1). Since this time is heavily system and hardware-dependent, it only serves as a reference point. Our times were measured on a Windows 11/Ubuntu 22.04 LTS dual-boot system with an Intel Core i9-10900K CPU. The macOS times were measured on a macOS Ventura system with an Apple M1 CPU.

## D  Keystroke Injections on Linux and macOS

### D.1  Keystroke Injections on Linux

On Linux systems, the available keyboard shortcuts and the general behavior of the UI depend on the *desktop environment*. While there are many different desktop environments and window managers, our work focuses on the three most widespread desktop environments: KDE Plasma, XFCE, and GNOME [80]. Here, our focus remains on recent versions, KDE Plasma 5, XFCE 4.0+, and GNOME 40+, that are shipped with many Linux distributions by default (e.g., Ubuntu/Kubuntu/Xubuntu 22.04 LTS).

**Run Dialogs.**  Multiple prominent run dialogs exist on KDE-, GNOME- and XFCE-based systems. By default, all three environments feature the same keyboard shortcuts to open terminals. Thus, Ctrl + Alt + T can be used in all environments to open the default terminal emulator with the default shell. Further, all environments feature a command window that can be used to enter arbitrary commands, which can be accessed using Alt + F2 in all three environments. Similar to Windows, pressing ⊞ provides a search in all three environments. This search can, however, only be used to open files or start applications. As such, it can open a terminal emulator, e.g., cmd. However, an attacker cannot obtain elevated privileges without additional effort, as all dialogs require a password. While elevated privileges granted to open applications might be reusable, we do not investigate this.

**Optimal Injections.**  The shortest total injection length amounts to 3 keystrokes: ⊞ + R , Ctrl + V , ↵ on our Linux sytems. Using the Keychron V1Z2, the entire injection only lasts about 5 ms. To our knowledge, this is also the fastest injection on Linux systems.

Table 1: **Tested devices and their macro functionality.** $\delta_{min}$ **is the average minimal time between individual keystrokes registered by the host and the corresponding standard error over** $1000$ **measurements. Devices with** *n/a* **in the WebHID column do not feature support for on-board macros.**

| | Vendor | Device | Connectivity | Firmware | Protocol | WebHID | Macro Length | $\delta_{min}$ in $ms$ |
|---|---|---|---|---|---|---|---|---|
| **MOUSE** | Logitech | G203 LIGHTSYNC | Wired | Logitech v152.2.17 | Logitech HID++ v4.2 | ✓ | ≈80 keys | 4.08 ± 0.01 |
| | | G305 LIGHTSPEED | Wireless Receiver | Logitech v68.1.14 | Logitech HID++ v4.2 | ✓ | ≈40 keys | 3.91 ± 0.02 |
| | | G500s | Wired | Logitech v84.9 | Logitech HID++ v1.0 | ✓ | ≈1076 keys | 7.94 ± 0.02 |
| | | G502 HERO | Wired | Logitech v127.3.10 | Logitech HID++ v4.2 | ✓ | ≈420 keys | 4.06 ± 0.01 |
| | Microsoft | Pro IntelliMouse | Wired | Microsoft 0095 | Custom HID | ✓ | 1 key | - |
| | Roccat | Kone Aimo | Wired | Roccat v1.05 | Custom HID | ✓ | >100 keys | 3.94 ± 0.01 |
| | Redragon | Pegasus M705 | Wired | ? | Custom HID | ✓ | 30 keys | 19.96 ± 0.02 |
| | Speedlink | TAUROX | Wired | ? | Custom HID | ✓ | 55 keys | 15.99 ± 0.02 |
| | Zelotes | T-90 | Wired | Gaming Mouse 3.0 | Custom HID | ✓ | ≈80 keys | 20.44 ± 0.04 |
| | Razer | Viper Ultimate | Wireless Receiver | Razer v1.06.00 | Custom USB | ✗ | >100 keys | 1.96 ± 0.00 |
| | Corsair | M55 RGB PRO | Wired | Corsair v4.7.23 | - | *n/a* | - | - |
| | SteelSeries | Rival 3 | Wired | SteelSeries 0.36.0.0 | - | *n/a* | - | - |
| | Asus | TUF Gaming M3 | Wired | Asus v1.00.09 | - | *n/a* | - | - |
| **KEYBOARD** | Logitech | G710+ Mechanical Keyboard | Wired | Logitech v0x8000 | - | *n/a* | - | - |
| | | MX Keys Mini | Bluetooth | ? | Logitech HID++ v4.2 | *n/a* | - | - |
| | Keychron | Keychron V1Z2 | Wired | VIA v3 | VIA Firmware Protocol v12 | ✓ | ≈408 keys | 1.94 ± 0.00 |
| | Skyloong | GK61XS | Wired | ? | Custom HID | ✓ | 30 keys | 17.65 ± 0.19 |
| | *Multiple* | 4 Key Macro Keypad | Wired | CH57x | Custom HID | ✓ | 5 keys | 48.00 ± 0.02 |
| | Redragon | K629 | Wired | ? | Custom HID | ✓ | 31 keys | 12.16 ± 0.14 |
| | Razer | BlackWidow 2019 | Wired | Razer v1.01.00 | Custom USB | ✗ | >100 keys | 1.95 ± 0.00 |
| **MISC.** | *Multiple* | USB Foot Switch FS221-P | Wired | FS22-P v5.3 | Custom HID | ✓ | 15 keys | 1.90 ± 0.01 |
| | Diswoe | Wireless Pro Controller | Bluetooth | ? | - | *n/a* | - | - |

Table 2: **Devices and their respective upper bound on the time to configure the shortest malicious payload.**

| | Vendor | Device | Time to Configure |
|---|---|---|---|
| **MOUSE** | Logitech | G203 LIGHTSYNC | ≈600 ms |
| | | G305 LIGHTSPEED | ≈600 ms |
| | | G500s LIGHTSYNC | ≈20 ms |
| | | G502 HERO | ≈600 ms |
| | Zelotes | T-90 | ≈180 ms |
| **KEYBOARD** | Keychron | V1Z2 | ≈55 ms |
| | *Multiple* | 4 Key Macro Keypad | ≈125 ms |

|  | 0 | 1 | 2 | 3 | 4 | 5 | 6 |
|---|---|---|---|---|---|---|---|
| | 0x18 | cmd | index | offset | page | data | end |

Figure 2: **Zelotes custom HID protocol.**

## D.2 Keystroke Injections on Apple's macOS

Apple's desktop OSs account for about 17 % of the respective market share [81]. Out of all versions, macOS Catalina is the most prominent one with above 91 % of adoption.

**Run Dialogs.** By default, macOS does not offer as many run dialogs as the other OSs. The most prominent run dialog is the Spotlight Search, where files, settings, and applications can be searched and opened. As such, it can be leveraged to open a terminal with the system's default shell. The search term may be shortened. If a terminal is already open, that terminal is brought to the foreground and into focus. Note that this also allows reusing permissions should the user have already entered a privileged state with the open terminal (e.g., using sudo).

**Optimal Injections.** With the shortest ephemeral payload being ⌘ + V , ↵ the shortest total injection length amounts to 12 keystrokes: ⌘ + [ ], T , E , R , M , I , N , A , L , ↵ , Ctrl

+ V , ↵ . Such an injection can be performed using most programmable devices discussed in Section 5, except for the macropad which can only store 5 keystrokes. Using the Keychron V1Z2, the entire injection only lasts about 125 ms. The injection requires two small delays: ≈10 ms until the search responsive and ≈110 ms until the terminal application becomes responsive.

## E Extension-based Mitigation

Listing 2 in the appendix shows an example of virtual machine layering. It replaces the functions sendReport and sendFeatureReport of any HID device with placeholders that do nothing. This effectively prevents any HID-based reprogramming of the device from WebHID. Alternatively, they could also be replaced by filter functions that act as a firewall. Freezing the device objects prevents the untrusted code from removing the encapsulations. Further, we can ensure that the scripts are injected into the site before executing any other script using the manifest in Listing 1. We run a content script before the rendering of any context starts such that no (potentially malicious) code can interfere with our encapsulation process.

## F Reverse-Engineered Protocols

**Zelotes.** The device communicates via HID feature reports with report ID '7'. All reports have the same length of 7 B. Figure 2 shows the report format. The protocol sends the binary internal state, consisting of button assignments, profiles, LED mode, and macro sequences, in blocks of 8 bytes, where each byte is encoded

Table 3: **Keystroke sequences to access run dialogs and the delays required for using them.**

| | Run Dialog Description | Keystrokes | Clipboard Content | Time to Payload |
|---|---|---|---|---|
| **Windows** | Start menu | ⊞ | | 60 ms |
| | Run command window | ⊞ + R | | 30 ms |
| | Quick Link menu into Run command window | ⊞ + X , R | | 110 ms |
| | Quick Link menu into Powershell | ⊞ + X , I | | 450 ms |
| | Quick Link menu into Powershell as Administrator | ⊞ + X , A , ◄ , ↵ | | 950 ms |
| **Linux** | Command window | Alt + F2 | | 0 ms |
| | Terminal | Ctrl + Alt + T | | 200 ms |
| | Search into Terminal | ⊞ , C , M , D , ↵ | | 550 ms |
| | | ⊞ , Ctrl + V , ↵ | cmd or terminal | 600 ms |
| **MacOS** | Spotlight Search into Terminal | ⌘ + Space , T , E , R , M , I , N , A , L , ↵ | | 120 ms |
| | | ⌘ + Space , ⌘ + V , ↵ | terminal | 90 ms |

```
 1  {
 2    "manifest_version": 2,
 3    "name": "Device Driver",
 4    "version": "1.0",
 5    "content_scripts": [
 6      {
 7        "matches": ["<all_urls>"],
 8        "js": ["sdk.js"],
 9        "run_at": "document_start",
10        "all_frames": true
11      }
12    ]
13  }
```

**Listing 1: This manifest ensures the content script is executed before the rendering of the document starts. It allows injecting the encapsulation function shown in Listing 2 which must run in the context of the webpage.**

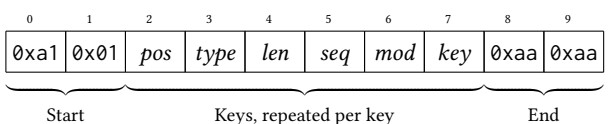

| 0 | 1 | 2 | 3 | 4 | 5 | 6 | 7 | 8 | 9 |
|---|---|---|---|---|---|---|---|---|---|
| 0xa1 | 0x01 | pos | type | len | seq | mod | key | 0xaa | 0xaa |

Start — Keys, repeated per key — End

Figure 3: **CH57x custom key-programming protocol. *pos* is the key position on the keyboard, *type* the type of key (normal, media, mouse), *len* the length of the macro sequence, *seq* the index of the key in the macro sequence, *mod* a bitfield for modifier keys, and *key* the HID code of the key.**

```
 1  (function () {
 2
 3  // originals are only accessible in this scope
 4  let original_requestDevice
 5    = window.navigator.hid.requestDevice;
 6  let original_getDevices
 7    = window.navigator.hid.getDevices;
 8
 9  let encapsulate = function (original) {
10    return async function () {
11      let devices = await original.apply(
12        window.navigator.hid, arguments);
13      // ... replace device.sendReport and
14      // device.sendFeatureReport on sensitive
15      // devices by encapsulations
16      for (let device of devices) {
17        device.sendReport = function () {
18          return;
19        }
20        device.sendFeatureReport = function () {
21          return;
22        }
23        // prevent removing encapsulations
24        // because it then returns to default
25        Object.freeze(device);
26      }
27      return devices;
28    };
29  };
30
31  window.navigator.hid.requestDevice
32    = encapsulate(original_requestDevice);
33  window.navigator.hid.getDevices
34    = encapsulate(original_getDevices);
35
36  })();
37
```

**Listing 2: This snippet removes the `sendReport` and `sendFeatureReport` function from all HID devices by encapsulating the original functions for device access. It has to be executed before any untrusted code.**

in a report. Reports use the command '0x3' to write a byte directly to the flash memory of the mouse. The location is calculated as page × 256 + offset + index (cf. Figure 2). After a maximum of 8 bytes, a block is "committed" using commands '0x9' and '0x0'. Command '0x5' finishes the button assignment, and command '0x10' indicates that the entire programming process has been completed.

**CH57x.** The communication with the device is via HID output reports with report ID '3'. All reports sent are 64 B, with unused bytes set to '0'. To program keys, the device first expects a "handshake" report, which consists of only '0's. After this report, the keys can be programmed. The basic protocol for programming keys is illustrated in Figure 3. The device expects a 2-byte start token (0xa1, 0x01) to start the programming mode. In programming mode, keys are sent one by one as 6-byte reports. Every report contains the position of the key on the keyboard, the type of key (e.g., normal key ('1'), media key ('2'), or mouse button ('3')), the number of keys in the macro sequence, the index within the macro sequence, and the key with one or more modifier keys (e.g., shift or control). A 2-byte end token (0xaaaa) terminates the programming mode.

