# OpenReview forum: "Peripheral Instinct: How External Devices Breach Browser Sandboxes"
_ACM.org/TheWebConf/2025/Conference — WWW 2025 Oral_

### Official Review · Reviewer_cUXT · 2024-11-02

**Novelty:** 3
**Technical Quality:** 3

**Review:**

Thanks for your work.

1. I think you need to improve your writing and presentation a lot.

2. You should back your claims by adding references. I have seen many claims without references. For example, “To facilitate these tasks, browsers introduced a range of APIs allowing websites to access various functionalities” here you can use a reference.

3. Your introduction section does not have a good flow. You put lots of information in the introduction. I think you should not mention prevention techniques here. Intro should have flow and need to show the importance and impact of the attacks.

4. Your three bullet contributions do not have anything. You just gave one line of descriptions for each bullet. I would suggest you write contributions consisting of the design process of attacks, evaluation process, and results showing some metrics (numbers). You may want to move some text from introduction to contribution.

5. I think you should add more references to your claims on the background section. For example, “Universal Serial Bus (USB) is a wired communication protocol for high-speed data transfer and power delivery. It is the de facto standard for connecting peripherals to computers” you should add reference for this line in the background.

6. “It can operate over various transport protocols, including USB and Bluetooth”... add reference.

7. “, while the W3C acknowledges that an HID “may contain [...] programmable macros” and suggests that “device manufacturers must [...] prevent a malicious app from reprogramming the device””...... add reference.

8. You sometimes come up with short terms like OS but never explained before like Operating System (OS). I would suggest you mention the term first and then you can write. Similar for AT, PWA.

9. You have referred to Appendix A but unfortunately you don't have Appendix. Yes, you have put some content after references but you should start these contents with a Section title Appendix and then you should refer to it.

10. “Given that prior work on browser extensions [69] and Android permissions [70] established that users rarely understand the risks associated with permissions, this likely also applies to browser APIs.”.... At least explain a little bit about these two works.

11. I think you should have something like a related study, or discussion section that explains and compares your work with existing work. I did not find any discussion showing your work's novelty from existing work. Whereas it seems most of your attacks are from existing work. Yes, the application of attacks in some cases may be emerging.

**Questions:**

1. What is the full form of AT command? You should mention it once and then you may use the short form.

2. Would you please tell me how your work is novel from existing work? Give me a detailed explanation please.

**Reviewer Confidence:**

3: The reviewer is confident but not certain that the evaluation is correct

**Scope:**

4: The work is relevant to the Web and to the track, and is of broad interest to the community

---

### Official Review · Reviewer_jRaD · 2024-11-24

**Novelty:** 3
**Technical Quality:** 3

**Review:**

This paper discusses the security and privacy problems caused by the rise of Web APIs. These APIs let websites directly connect to external devices, changing how devices are trusted and creating new risks. The study shows that attackers can use these APIs to fully control devices and even bypass browser protections to run any code they want.

Strengths:
- The paper uses various methods, including multi-device testing and attack chain construction, to confirm the theoretical feasibility of attacks on Web APIs.
- The paper proposes several measures to mitigate attacks on Web APIs, which have received positive feedback from some device vendors.

Weakness:
- The paper mainly relies on lengthy text and lacks necessary illustrations or diagrams.
- The chapter structure of this paper could be improved. It is recommended to organize the chapters into clear categories or provide a summary of each chapter or group of chapters in the Introduction.

**Questions:**

- Can you summarize the experimental data in a table after discussing the attacks on Web APIs?
- Can you explain the practicality, cost, and impact on the system of each suggested mitigation for Web API attacks?
- Can you explain whether similar issues exist in non-Chromium browsers?

**Reviewer Confidence:**

3: The reviewer is confident but not certain that the evaluation is correct

**Scope:**

4: The work is relevant to the Web and to the track, and is of broad interest to the community

---

### Official Review · Reviewer_9z4j · 2024-11-27

**Novelty:** 3
**Technical Quality:** 3

**Review:**

The paper explores and shows the risks associated with browser APIs such as WebHID, WebUSB, Web Serial, and WebMIDI. These APIs can interact with physical devices connected to the host machine, potentially exploiting them to gain persistent access to both the devices and the host. Despite the need to explicitly authorize a website before it can use the APIs, the authors claim that this protection is not enough to protect from these attacks since malicious websites can impersonate trusted vendors. Furthermore, once these permissions are given, a user needs to revoke them explicitly, they will not expire once the tab or browser session is closed.
The paper also presents possible attack scenarios, including firmware attacks that can be carried out against the devices using the APIs. But also exploiting device-specific protocols and functionalities, such as macros in keyboards, to achieve command injection or install spyware. In its last part, the paper presents possible mitigation strategies against the attacks that were previously surveyed. The protections are divided into various approaches: firewall-like solutions, browser-based, device-based, and host-based mitigations.

The paper is well-written and clear. The paper's novelty mainly resides in exploring the possible risks of browser APIs communicating with physical devices. In this sense, the paper does a good job surveying the security issues behind them. The topic has a practical impact despite the current availability of APIs that are still limited to Chrome and Edge browsers.

The main issue in this work is the lack of a scientific or systematic methodology supporting it. If the authors used such a methodology, it is not presented in the paper, which is structured to present a list of attacks and potential defences.
Another suggestion to improve the paper is related to sections 4 to 7, where various attacks on the devices are described. These sections focus on which exploits can be done on the devices. Most of the exploits shown are widely known, such as firmware attacks such as replacement or rollbacks. Furthermore, these sections shift the paper's focus from the Browser API dangers to a survey of device-related attacks, making the scope of the paper less relevant to the web and more related to device exploitation.

The following is a summary of the pros and cons of this work.

Pros:
- Well written, well-explained, and clear
- Contributes to raising awareness on the dangers of browser APIs that communicate with physical devices
- Shows a set of possible defences against the attacks

Cons:
- Lack of a methodology for evaluating and analyzing browser APIs' security problems and their impact.
- The authors extensively describe device-specific attacks that are already known, which greatly weakens the paper’s relevance to the web and its overall novelty.
- A long part of the paper (sections from 4 to 7) describes known device-related exploits rather than focusing on the web-related part of the attacks.

**Questions:**

Is there a methodology or systematic evaluation that you followed to analyze the attacks and defences you propose?


Is there a novelty in the device-related attacks you describe, except for the fact that they were delivered through the browser APIs?

**Reviewer Confidence:**

3: The reviewer is confident but not certain that the evaluation is correct

**Scope:**

2: The connection to the Web is incidental, e.g., use of Web data or API

---

### Official Review · Reviewer_2MFU · 2024-11-29

**Novelty:** 3
**Technical Quality:** 3

**Review:**

The paper studies the security mechanism of web browsers' API regarding the interaction with external peripheral devices. Along with the evolution of web technologies, peripheral devices can bypass the centralized control at the OS level and be directly utilized by the web browser, which may introduce potential attack surfaces when the browser hosts the malicious website. Based on this insight, a comprehensive and thorough analysis of the browser APIs was conducted, and some attack methodologies have been proposed. In the end, the author also discussed some actionable mitigating strategies to foster trust in the web usage of peripheral devices. The paper studies a very critical problem that concerns web security and the studied area is closely related to the scope of the WebConf. However, the paper is overall poorly written and hard to follow. The organization and writing of the paper need to be significantly improved to make the reading a fluent and comfortable process.

**Questions:**

* *Introduction & background*. The introduction is not well self-explainable. Until the end of the third paragraph, the paper has still not explained “*why the host is shifted from OS to the website?*” and *“What is the case that the website as the host is not trusted and has changed the threat model?*” In other words, *what exactly motivates this research?* And *why it is significant?* These questions should be explicitly addressed and presented in the Introduction. Either some references or concrete examples of incidents that have ever taken place (e.g., well-known security breaches) would be helpful for the reading. In addition, the background is also poorly organized. It looks like a rushed work without necessary revision and check. E.g., the shortwriting of HID is not given at its first occurrence, and the entire Sec 2.1 is poorly structured and hard to follow.

* *Completeness of browser APIs*. In section 2.2.1, the author claims “five device APIs are available”. Is there any evidence to back this claim? The reference, i.e., [20], does not mention there only exist five APIs. Do these five APIs represent all APIs for *peripheral devices*? How about WebNFC, WebVR/WebXR, and WebGPU?

* *Systemization of the knowledge*. One of the biggest issues of this paper is the poor systemization of knowledge, making it look more like a technical disclosure report for a training session rather than a research paper. For example, Section 4 introduces framework attacks, but the remaining two categories of attacks are not presented in the sequential sections but skipped to Sections 6 and 7. This can also be reflected in the attack sections, in which the author presents a lot of technically intensive content. Unfortunately, as the reader, I find it hard to catch the research contribution and construct the big picture. It is hard to connect and compare the six different types of mitigations presented in Section 8. It is also unclear whether the attacks and mitigations are complete.

**Reviewer Confidence:**

3: The reviewer is confident but not certain that the evaluation is correct

**Scope:**

4: The work is relevant to the Web and to the track, and is of broad interest to the community

---

### Official Review · Reviewer_ihpo · 2024-11-30

**Novelty:** 4
**Technical Quality:** 6

**Review:**

Thank you for submitting this work to WWW'25! The paper presents an approach to how new browser APIs can be used to abuse known vulnerabilities of peripheral devices. The authors show that the new APIs do not properly protect against the named known attack vectors.
The paper is well-written and easy to follow. The authors present their approach and results in a clear way, making it easy to assess the findings. This reviewer hopes that the following comments and questions help the authors to further improve their work.

[MAJOR] One of my main concerns with this work is the proposed assumption of the stack model that each peripheral device assumes that the host device can be trusted and that the new APIs change this assumption fundamentally. The host device is not trustworthy (e.g., it was infected with malware, or the device is accessed by an untrustworthy third-party tool) even when the new APIS is not present. Thus, the claim that the threat model has fundamentally changed is an exaggeration. This reviewer agrees that the attack surface increased.

[MINOR] The firmware-based attacks (Section 4) are only done for two types of (old) Logitech systems. Thus, it is not clear if and how the proposed attacks scale in the wild or if they are only applicable to old devices.

[MAJOR] The authors state that many manufacturers use device-specific protocols (i.e., not even manufacturer-specific). Thus, a generalizability question arises regarding whether the results presented scale to many other devices. Further, it is unclear how much work it would be for an adversary to craft an exploit that would work for many devices. This fact should be addressed in the threat model.

Overall, the paper presents an interesting new perspective on newly emerging threats that new APIs introduce. The authors show that classical sandboxing concepts can be bypassed undermining fundamental security concepts. The results show that browser vendors need to closely consider security aspects when building and shipping new features. However, the proposed paper mainly relies on existing and well-documented threats and shows that they are abusable from within the browser.

**Questions:**

See the [MAJOR] comments above.

**Reviewer Confidence:**

3: The reviewer is confident but not certain that the evaluation is correct

**Scope:**

3: The work is somewhat relevant to the Web and to the track, and is of narrow interest to a sub-community